# Nitric Oxide Linked to mGluR5 Upregulates BDNF Synthesis by Activating MMP2 in the Caudate and Putamen after Challenge Exposure to Nicotine in Rats

**DOI:** 10.3390/ijms231810950

**Published:** 2022-09-19

**Authors:** Jieun Kim, Sumin Sohn, Sunghyun Kim, Eun Sang Choe

**Affiliations:** Department of Biological Sciences, Pusan National University, 63-2 Busandaehak-ro, Gumjeong-gu, Busan 46241, Korea

**Keywords:** endopeptidase, glutamate receptor, neurotropic factor, nitric oxide, striatum

## Abstract

Nitric oxide (NO) linked to glutamate receptors in the caudate and putamen (CPu) regulates neuroadaptation after drug exposure. Matrix-metalloproteinase (MMP), a Ca^2+^-dependent zinc-containing endopeptidase, increases mature brain-derived neurotrophic factor (BDNF) synthesis after drug exposure in the brain. The present study determined that NO synthesis linked to metabotropic glutamate receptor subtype 5 (mGluR5) stimulation after challenge exposure to nicotine activates MMP, which upregulates BDNF synthesis in the CPu. Subcutaneous injection of challenge nicotine (1.0 mg/kg) after repeated injections of nicotine (1.0 mg/kg/day) for 14 days and 7 days of nicotine withdrawal increased MMP2 activity and BDNF expression in the CPu of rats. These increases were prevented by the bilateral intra-CPu infusion of the mGluR5 antagonist, MPEP (0.1 nmol/side), the IP_3_ receptor antagonist, xestospongin C (0.004 nmol/side) or the neuronal nitric oxide synthase (nNOS) and NO inhibitor, Nω-propyl (0.1 nmol/side) prior to the challenge nicotine. Furthermore, bilateral intra-CPu infusion of the MMP2 inhibitor, OA-Hy (1 nmol/side) prevented the challenge nicotine-induced increase in the expression of BDNF. These findings suggest that elevation of NO synthesis linked to mGluR5 potentiates BDNF synthesis via activation of MMP2 after challenge exposure to nicotine in the CPu of rats.

## 1. Introduction

Nicotine, a major psychoactive component of tobacco, causes compulsive tobacco smoking by upregulating glutamate release in the brain [1,2,3]. Stimulation of α7 nicotinic acetylcholine receptors (nAChRs) that are expressed mainly in the terminals of the caudate and putamen (CPu) projected from prefrontal cortex enhances glutamate release after nicotine exposure [4]. Exposure to challenge nicotine stimulates α7 nAChRs in the CPu of rats, while blockade of these receptors decreases the challenge nicotine-induced increase in glutamate concentration in the CPu of rats [4].

Elevation of glutamate release stimulates metabotropic glutamate receptor subtype 5 (mGluR5), which is expressed dominantly in the CPu [5,6,7]. Stimulation of the mGluR5 after nicotine exposure leads to intracellular Ca^2+^ mobilization by increasing the concentrations of diacylglycerol (DAG) and inositol trisphosphate (IP_3_) as a result of Gαq-stimulated phosphoinositide (PI) hydrolysis in the NAc [8]. Elevated Ca^2+^ mobilization increases nitric oxide (NO) efflux by changing the phosphorylation state of neuronal nitric oxide synthase (nNOS) after cocaine exposure in the CPu [9]. These findings suggest that challenge exposure to nicotine leads to an increase in glutamate release, which has the capability to evoke NO synthesis by stimulating mGluR5 in the CPu.

Matrix-metalloproteinase (MMP), a Ca^2+^-dependent zinc-containing endopeptidase belonging to the metzincin superfamily, cleaves other peptides proteolytically and regulates a variety of cellular functions including neuroadaptation [10,11,12]. MMP is initially synthesized as an inactive form, pro-peptide, which must be processed into active peptide to participate in catalytic activity [13]. MMP can be subdivided into gelatinase, collagenase, stromelysin, matrilysin, membrane-type MMP, and others [13]. MMP2 belonged to the gelatinase subfamily, called gelatinase A (72 kDa) regulates synaptic plasticity and physiology by processing extracellular matrix glycoproteins to initiate glutamate receptor trafficking and actin polymerization [13,14]. Previous studies have shown that nicotine self-administration increases MMP2 activity in the nucleus accumbens (NAc) of rats [10]. However, it is not known how they regulate neuronal plasticity in the CPu after challenge exposure to nicotine.

Brain-derived neurotrophic factor (BDNF), a family of neurotrophin polypeptide including nerve growth factor and neurotrophins, is expressed in the nervous system [15,16]. BDNF is constitutively synthesized as a pro-BDNF (32 kDa) that is processed to a smaller mature form by endopeptidases, such as the Golgi-resident- and secretory granule-resident proprotein convertase 1–7, and the membrane-anchored protein, furin, through the trans-Golgi network [15]. Cleaved BDNF is further processed to mature BDNF (BDNF) by endopeptidases, such as MMP and tissue-plasminogen activator that are located outside of neurons. BDNF then upregulates the functions of GABAergic neurons in the CPu by stimulating its receptor, BDNF-activated tropomyosin receptor kinase B (TrkB) [15,17]. Previous studies have demonstrated that nicotine upregulates BDNF expression in the GABAergic neurons of the striatum [18,19]. Stimulation of mGluR5 increases BDNF expression in cultured C6 glioma cells [20]. Taken all together, activation of MMP2 after nicotine exposure may be necessary for synthesizing BDNF probably via mGluR5 stimulation in the CPu.

Therefore, the present study was undertaken to determine how NO synthesis linked to mGluR5 stimulation after challenge exposure to nicotine interacts with MMP and upregulates BDNF synthesis in the CPu, a component of the basal ganglia involving habit learning such as cigarette smoking.

## 2. Results

### 2.1. Challenge Nicotine Increased MMP2 Activity and BDNF Expression in the CPu of Rats

This study determined whether the subcutaneous injection of challenge nicotine (1.0 mg/kg) alters MMP2 activity and BDNF expression in the CPu of rats using ELISA, qPCR, and Western blotting analyses (Figure 1A). Challenge nicotine after repeated injections of nicotine for 14 consecutive days (1.0 mg/kg/day) and 6 days of withdrawal significantly increased MMP2 activity (Figure 1B, t(4) = 4.194, df = 8, *p* = 0.003, unpaired *t*-test). Challenge nicotine also increased the levels of total BDNF and BDNF protein expression, but not *bdnf* mRNA transcription (Figure 1C–E: C, t(3) = 3.589, df = 6, *p* = 0.0115, unpaired *t*-test; D, t(8) = 8.084, df = 4, *p* = 0.0013, unpaired *t*-test; E, t(0) = 0.2644, df = 6, *p* = 0.8003, unpaired *t*-test).

### 2.2. Blockade of mGluR5 Decreased the Challenge Nicotine-Induced Increases in MMP2 and BDNF Levels

Since challenge exposure to nicotine increased MMP2 activity and BDNF expression in the CPu, this study determined whether they are linked to mGluR5 stimulation after challenge nicotine. Systemic challenge nicotine (1.0 mg/kg) decreased gelatin-conjugated fluorescein isothiocyanate (FITC) fluorescence intensity. However, bilateral intra-CPu infusion of the mGluR5 antagonist, MPEP (0.1 nmol/side) did not (Figure 2B). Bilateral intra-CPu infusion of MPEP significantly decreased the challenge nicotine-induced increase in MMP2 activity (Figure 2D, interaction, F(1,16)  =  17.62, *p* = 0.0007; drug effect, F(1,16)  =  47.67, *p*  < 0.0001; nicotine effect, F(1,16)  =  12.48, *p*  = 0.0028, two-way ANOVA), and total BDNF expression (Figure 2F, F(3,4)  =  2.727, F(3,4)  =  15.71, unpaired *t*-test). Bilateral intra-CPu infusion of MPEP significantly decreased the co-localization of mGluR5 and BDNF in neurons of the CPu, which was elevated by challenge nicotine when compared to challenge saline control (Figure 2G).

### 2.3. Blockade of mGluR5 Decreased the Challenge Nicotine-Induced Increases in IP_3_, Phosphorylated (p)-nNOS and NO Levels

Challenge exposure to nicotine stimulates mGluR5 by increasing extracellular glutamate concentration in the CPu [4,8]. Stimulation of mGluR5 upregulates NO synthesis in the CPu by increasing IP_3_ concentration after drug exposure [9,21]. Based on the evidence, this study determined whether stimulation of mGluR5 alters activation of nNOS and NO synthesis in association with IP_3_ production in the CPu after challenge exposure to nicotine. Bilateral intra-CPu infusion of the mGluR5 antagonist, MPEP significantly decreased the challenge nicotine-induced increases in the levels of IP_3_ (Figure 3A, interaction, F(1,31)  =  11.35, *p*  = 0.0020; drug effect, F(1,31)  =  8.67, *p*  = 0.0061; nicotine effect, F(1,31)  =  2.338, *p*  = 0.1364, two-way ANOVA), *p*-nNOS (Figure 3B, interaction, F(1,10)  =  8.068, *p*  = 0.0175; drug effect, F(1,10)  =  4.134, *p*  = 0.0694; nicotine effect, F(1,16)  =  25.76, *p*  = 0.0005, two-way ANOVA), and NO (Figure 3C, interaction, F(1,8) = 9.662, *p* = 0.0145; drug effect, F(1,8) = 5.306, *p* = 0.0502; nicotine effect, F(1,8) = 8.767, *p* = 0.0181, two-way ANOVA).

### 2.4. Blockade of IP_3_ Receptors Decreased the Challenge Nicotine-Induced Increase in the Levels of p-nNOS, NO, MMP2, and BDNF

Since challenge nicotine increased IP_3_ and nNOS/NO levels by stimulating mGluR5, this study determined whether stimulation of IP_3_ receptors that are coupled to mGluR5 alters MMP2 activity and BDNF expression by increasing NO synthesis following nNOS activity in the CPu. Bilateral intra-CPu infusion of the IP_3_ receptor antagonist, xestospongin C (0.004 nmol/side) significantly decreased the challenge nicotine-induced increases in the levels of p-nNOS (Figure 3E, interaction, F(1,10)  =  1.049, *p*  = 0.3299; drug effect, F(1,10)  =  3.129, *p*  = 0.1073; nicotine effect, F(1,10)  =  2.338, *p*  = 0.0105, two-way ANOVA), NO (Figure 3F, interaction, F(1,12)  =  10.67, *p*  = 0.0067; drug effect, F(1,12)  =  1.714, *p*  = 0.2150; nicotine effect, F(1,12)  =  2.52, *p*  = 0.1384, two-way ANOVA), MMP2 activity (Figure 3G, interaction, F(1,16)  =  38.62, *p*  < 0.0001; drug effect, F(1,16)  =  98.07, *p*  = 0.0001; nicotine effect, F(1,16)  =  22.42, *p*  = 0.0002, Two-way ANOVA), and BDNF (Figure 3H, interaction, F(1,9)  =  3.311, *p*  = 0.01021; drug effect, F(1,9)  =  6.444, *p*  = 0.0318; nicotine effect, F(1,9)  =  3.658, *p*  = 0.0881, two-way ANOVA).

### 2.5. Inhibition of nNOS Decreased the Challenge Nicotine-Induced Increase in the Levels of NO, MMP2 and BDNF

Since IP_3_ receptors linked to mGluR5 upregulate NO synthesis, this study determined whether NO regulates MMP2 activity and BDNF expression after challenge exposure to nicotine. Bilateral intra-CPu infusion of the nNOS inhibitor, Nω-propyl (0.1 nmol/side) significantly decreased the levels of NO (Figure 4B, interaction, F(1,12) = 7.314, *p* = 0.0192; drug effect, F(1,12) = 5.207, *p* = 0.0415, nicotine effect, F(1,12) = 6.272, *p* = 0.0277, two-way ANOVA), and MMP2 activity (Figure 4C, interaction, F(1,16)  =  14.48, *p*  = 0.0016; drug effect, F(1,16)  =  53.95, *p* < 0.0001; nicotine effect, F(1,16)  =  6.984, *p* = 0.0177, two-way ANOVA), and BDNF (Figure 4D, interaction, F(1,11) = 4.029, *p* = 0.0699; drug effect, F(1,11) = 27.27, *p* = 0.0003, nicotine effect, F(1,11) = 0.4331, *p* = 0.5240, two-way ANOVA) in the CPu which were elevated by challenge nicotine.

### 2.6. Inhibition of MMP2 Decreased the Challenge Nicotine-Induced Increase in the Level of BDNF

Since NO regulates MMP2 activity and BDNF expression, this study determined whether MMP2 activity is directly associated with BDNF expression after challenge nicotine. Bilateral intra-CPu infusion of the nNOS inhibitor, Nω-propyl (0.1 nmol/side) significantly decreased the level of BDNF in the CPu which was elevated by challenge nicotine (Figure 4D). Similar results were obtained by the bilateral intra-CPu infusion of the MMP2 inhibitor, OA-Hy (1 nmol/side) (Figure 4E, interaction, F(1,10) = 3.71, *p* = 0.0830; drug effect, F(1,10) = 27.69, *p* = 0.0004, nicotine effect, F(1,10) = 0.09487, *p* = 0.7644, two-way ANOVA). Throughout the experiments, bilateral intra-CPu injections of the inhibitors prior to challenge nicotine were mostly located in the center of the CPu.

## 3. Discussion

The present results demonstrate that elevation of NO synthesis after challenge exposure to nicotine activates MMP2, resulting in the upregulation of BDNF synthesis via mGluR5 stimulation in the CPu of rats. Previous study has demonstrated that nicotine increases MMP2 activity in the NAc of rats [10]. Nicotine also increases BDNF expression in the glutamate synapses of the striatum in rats and mice [19,22]. It is well-known that challenge exposure to nicotine increases extracellular glutamate concentration in the CPu of rats [4]. Taken together, these findings suggest that challenge exposure to nicotine increases MMP2 activity and BDNF synthesis by increasing glutamate release in the CPu.

In this study, the blockade of mGluR5 prevents MMP2 activity and BDNF expression that are elevated by challenge exposure to nicotine. It is known that mGluR5 is expressed dominantly in the CPu [23]. Repeated exposure to nicotine increases the surface expression of mGluR5 as well as glutamate concentration in the CPu [8,24]. Intra-NAc infusion of FITC-conjugated gelatin peptide increases the FITC fluorescence after cocaine reinstatement in the NAc of rats [25], supporting the possibility that challenge exposure to nicotine has potential for activating MMP2 in the CPu. In the present study, active MMP2 induced by challenge exposure to nicotine cleaves FITC-conjugated gelatin peptide, but not the intra-CPu infusion of the selective mGluR5 antagonist, MPEP prior to challenge nicotine. These data support our finding that elevation of MMP2 activity and BDNF synthesis after challenge exposure to nicotine may be the consequence of stimulating mGluR5 in GABAergic neurons of the CPu. However, it is possible that stimulation of astroglial mGluR5 may contribute to BDNF synthesis by increasing glutamate release in the CPu after challenge exposure to nicotine [26,27].

Blockade of mGluR5 prevents the challenge nicotine-induced elevation of IP_3_, p-nNOS, and NO concentrations in this study. Parallel to our findings, previous studies have shown that mGluR5 coupled to Gαq in GABAergic neurons increases intracellular Ca^2+^ mobilization by activating the PLC-mediated DAG and IP_3_ pathways [21]. Blockade of mGluR5 decreases intracellular Ca^2+^ concentration by decreasing IP_3_ production that is elevated by nicotine administration in the rat NAc [8]. Furthermore, blockade of mGluR5 with MTEP, a mGluR5 negative allosteric modulator, downregulates the glutamate-induced increase in the levels of NO in the NAc [25]. Inhibition of nNOS also decreased mGluR5 upregulation-induced increase in the levels of NO [25]. These findings suggest that challenge exposure to nicotine leads to an increase in NO synthesis by stimulating mGluR5-coupled IP_3_ production in the CPu. This speculation can be supported by the finding that blockade of IP_3_ receptor reduces the levels of p-nNOS and NO concentrations as demonstrated in this study. Furthermore, blockade of IP_3_ receptors decreases the elevation of MMP2 activity and BDNF expression caused by challenge exposure to nicotine in this study. Taken together, these findings suggest that elevation of nNOS activation followed by NO synthesis by stimulating the IP_3_ receptor after challenge exposure to nicotine is linked to mGluR5 stimulation. This leads to MMP2 activation and results in the elevation of BDNF synthesis in the CPu.

In the present study, we demonstrated that inhibition of nNOS decreases the challenge nicotine-induced increase in NO, MMP2 activity, and BDNF expression in the CPu of rats. MMP2 is activated by S-nitrosylation in the extracellular space after cocaine exposure [25]. Inhibition of nNOS in the NAc decreases MMP2 activity which is elevated by cocaine self-administration in rats [25]. Chemogenetic stimulation of nNOS-expressing interneurons increases MMP2 activity, which is abolished by a nNOS inhibitor in the NAc of mice after cocaine self-administration [25]. Taken together, these findings suggest that activation of nNOS followed by NO synthesis is linked to mGluR5 after challenge exposure to nicotine converts inactive MMP2 to active molecule, which results in the upregulation of BDNF synthesis in the CPu. This prediction can be supported by the present finding that inhibition of MMP2 decreases the challenge nicotine-induced increase in the expression of BDNF in the CPu. Taken together, these findings suggest that activation of MMP2 by elevated NO synthesis is required for BDNF synthesis in the CPu after challenge exposure to nicotine in rats.

## 4. Materials and Methods

### 4.1. Animals

Male Sprague-Dawley rats (6 weeks old, weight 210–250 g) were purchased from Hyo-Chang Science Co. (Daegu, Korea). The rats were housed in pairs in a controlled environment under a 12-h light–dark cycle. Temperature and humidity were maintained at 21–23 °C and 45–55%, respectively. Food and water were provided ad libitum. The rats were allowed to acclimate for a minimum of 5 days before conducting experiments. Experiments were performed in a quiet room to reduce stress of the animals. All animal procedures were approved by the Institutional Animal Care and Use Committee of Pusan National University and conducted in accordance with the provisions of the NIH Guide for the Care and Use of Laboratory Animals.

### 4.2. Drugs

All pharmacological drugs, except nicotine (Sigma-Aldrich, St. Louis, MO, USA), were purchased from Tocris Bioscience (Bristol, UK). Working solutions were always prepared freshly for experiments. The non-competitive mGluR5 antagonist, MPEP (0.5 nmol), the IP_3_ receptor antagonist, xestospongin C (0.004 nmol), the nNOS and NO inhibitor, Nω-propyl (0.1 nmol), and the MMP2 inhibitor, cis-9-octadecenoyl-N-hydroxylamide (OA-Hy, 0.1 nmol), were dissolved in the minimum concentration of dimethyl sulfoxide (DMSO) and then diluted in artificial cerebro-spinal fluid (aCSF) containing (mM) 123 NaCl, 0.86 CaCl_2_, 3.0 KCl, 0.89 MgCl_2_, 0.50 NaH_2_PO_4_, adjusted to neutral with NaCl. The same DMSO/aCSF mixture solution was used as a vehicle control for a given drug. The drug solutions were all adjusted to neutral with NaOH if necessary. Rats received repeated subcutaneous injections of saline or nicotine (1.0 mg/kg) once a day for 14 consecutive days. Challenge nicotine was injected as 1.0 mg/kg of rats and saline was used as a vehicle after 6 days of withdrawal (Figure 1A). Nicotine was dissolved in physiological saline (0.9% sodium chloride). The concentrations of drugs were determined from previous studies [28,29].

### 4.3. Surgery and Intra-CPu Drug Infusion

Rats were anesthetized with the mixture of Zoletil 50 (75 μL/kg) (Virbac Korea, Seoul, Korea) and Rompun (50 μL/kg) (Bayer Korea, Seoul, Korea) through subcutaneous injections and were then placed in a stereotaxic apparatus. Under aseptic conditions, a 23-gauge stainless steel guide cannula (0.29 mm inner diameter, 10 mm in length) was implanted 1 mm anterior to the bregma, 2.5 mm left and right of the midline, and 5 mm below the surface of the skull. The guide cannula was sealed with a stainless steel wire of the same length. The rats were then allowed to recover from surgery for 5 days prior to the experiment. On the day of the experiment, the inner steel wire was replaced with a 30-gauge stainless steel injection cannula (0.15 mm inner diameter, 12.5 mm in length) that protruded 0.5 mm beyond the guide cannula. Throughout the experiments, all drugs and FITC-conjugated gelatin peptide were infused bilaterally into the central part of the CPu 5 min prior to the final injection of saline or nicotine in a volume of 1 μL at a rate of 0.2 μL/min in freely moving rats (Figure 2A). The progress of the injection was monitored by observing the movement of a small air bubble along the length of precalibrated PE-10 tubing inserted between the injection cannula and a 2.5 μL Hamilton microsyringe (Fisher Scientific, Pittsburgh, PA, USA). After completing the injection, the injector was left in place for an additional 5 min to reduce any possible backflow of the drug along the injection tract. The physical accuracy of the injection was verified by the reconstruction of microinjection placements (Figure 2, Figure 3 and Figure 4). Gliosis caused by the implantation of the guide cannula and the infusion of drugs was not detected by Nissl staining (data not shown).

### 4.4. Enzyme-Linked Immunosorbent Assay (ELISA)

ELISA was performed as the recommended procedure in total BDNF sandwich ELISA kit (Millipore, Darmstadt, Germany). Briefly, rats were deeply anesthetized with the mixture of Zoletil 50 and Rompun, then decapitated 30 min after the final drug injection. Next, brains were removed, frozen in isopentane at −70 °C, and stored in a deep freezer until use. Brain sections were serially cut using a cryostat (Leica biosystems, Nussloch, Germany) at −25 °C, after which the injected both sides of the CPu were removed with a steel borer (2 mm inner diameter). All tissue samples were transferred to homogenization buffer containing 2% bovine serum albumin, 1 M NaCl, 4 mM EDTA, 2% Triton X-100, 0.1% sodium azide and the protease inhibitors (μg/mL) containing 5 aprotinin, 0.5 antipain, 157 benzamidine, 0.1 pepstain A, and 17 phenylmethyl-sulphonyl fluoride. Homogenates were prepared in 50 volumes of the homogenization buffer to tissue weight and sonicated three times for 9 s each and then incubated on ice for 1 h. After sonicating, samples were centrifuged at 14,000× *g* for 30 min at 4 °C. The pellet, which primarily contained nuclei and large debris, was discarded, and the supernatant was centrifuged again at the same condition. The supernatants were added 100 μL to each well and incubated overnight on a shaker at 4 °C. After incubating, the samples were removed and incubated with biotinylated mouse anti-BDNF for 3 h. After incubating, streptavidin-HRP conjugate solution was added and incubated for 1 h. Finally, TMB/E solutions were added to each well and shook for 15 min. The reaction was halted by adding stop solutions and the plate was then immediately read at 450 nm using microplate multi reader (Promega, Madison, WI, USA). All reagents were diluted using sample diluent (1:1000) in the kit and each well was washed at least three times between respective steps. Standard curves were generated using recombinant human BDNF for each experiment.

Cellular MMP2 activity was measured in rat CPu using a SensoLyte Plus MMP2 ELISA Kit from ANASPEC (Fremont, CA, USA) according to the manufacturer’s instruction. Cellular total NO synthesis was measured in rat CPu using a NO ELISA Kit from Abcam (Cambridge, UK) according to the manufacturer’s instruction. The IP_3_ production level was determined using an IP_3_ ELISA kit (Cusabio Biotech Co., Wuhan, China) according to the manufacturer’s instruction. Two freeze–thaw cycles were performed to break down cell membranes, and the homogenates were centrifuged for 5000× *g* at 5 min at 4 °C. The supernatant was removed and assayed immediately. An anti-IP_3_ detection antiserum was added and then incubated for 60 min at 37 °C, followed by the addition of the substrate solution for 15 min at 37 °C. The reaction was terminated following the addition of the stop solution, and the plates were read at 450 nm absorbance by a GloMax-Multi Microplate Multimode Reader (Promega).

### 4.5. Double Immunofluorescence Staining

Rats were deeply anesthetized with a mixture of Zoletil 50 and Rompun via subcutaneous injections and then transcardially perfused with 4% paraformaldehyde in PBS at 4 °C. The brains were then removed and post-fixed in a solution of 10% sucrose in 4% paraformaldehyde for 2 h at 4 °C, after which they were placed in 20% sucrose in PBS and held at 4 °C overnight [30]. Using a freezing sliding microtome, 16 μm frozen sections were obtained. Three sections per brain were used for the staining. The sections were blocked with a blocking buffer containing 4% normal goat serum and 1% bovine serum albumin in PBS, and then washed with PBS three times for 10 min each. The sections were then incubated in a mixture of rabbit or mouse antiserum against BDNF (Abcam) (1:100) and mGluR5 (Abcam) (1:500), respectively, for 72 h at 4 °C on a shaker. After rinsing the sections several times in PBS, the sections were incubated in a mixture of goat anti-mouse and rabbit secondary antisera conjugated to Alexa Fluor 488 and 594 (Abcam) (1:200) for 2 h. The sections were rinsed three times and mounted on slide glasses, cover-slipped, and monitored using a fluorescence camera (Carl Zeiss, Jena, Germany) with specific filters for detecting specific wavelengths. Images of the sections were obtained by using ZEN 2012 SP2 (blue edition) image processing software (V2.0 en, Carl Zeiss, Jena, Germany). The representative images are captured by ultra-high-speed and spectral confocal microscope (model number A1 Rsi/Ti-E, Nikon, Tokyo, Japan).

### 4.6. Western Immunoblotting

Western immunoblotting was performed as described previously [30]. All tissue sample was transferred to homogenization buffer containing (mM) 10 Tris-HCl, pH 7.4, 5 NaF, 1 Na_3_VO_4_, 1 EDTA, and 1 EGTA. Homogenates were sonicated three times for 9 sec each and then incubated on ice for 1 h. After sonicating, samples were centrifuged at 14,000× *g* for 30 min at 4 °C. The pellet, which primarily contains nuclei and large debris, was discarded, and the supernatant was centrifuged again at the same condition. The concentration of the solubilized proteins in the supernatant fraction was determined using a Bio-Rad Protein Assay (Bio-Rad Laboratories, Hercules, CA, USA). The proteins in the supernatant were resolved using 20% sodium dodecyl sulfate-polyacrylamide gel electrophoresis, after which the separated proteins were transferred to a nitrocellulose membrane. Next, the membrane was blocked with blocking buffer containing 5% skim milk in a mixture of tris buffered saline and tween-20 (TBST), then washed three times for 10 min each with TBST. After washing, the membrane was probed with either a rabbit primary antiserum against BDNF (Abcam) (1:1000) or phospho-nNOS (Abcam) (1:1000), for 18 h at 4 °C on a shaker. The membrane was washed again, then incubated with an appropriate rabbit secondary antiserum (KPL, Gaithersburg, MD, USA) (1:1000) for 1 h at room temperature. Next, immunoreactive protein bands were detected using iBright CL 1000 (Invitrogen, Waltham, CA, USA) enhanced chemi-luminescence reagents (Ab Frontier, Seoul, Korea). The same membrane was also probed for β-tubulin (Cell Signaling, Danvers, MA, USA) (1:2000) to normalize the blots. Either rabbit primary or secondary antiserum against β-tubulin was diluted to 1:1000. Immunoreactive protein bands visualized on membrane were semi-quantified using an imaging digital camera and the NIH Image 1.62 software (Bethesda, MD, USA) as described previously [30].

### 4.7. Statistics

Differences in the number of immunoreactive pixels per measured area from Western immunoblotting, and ELISA between groups were determined by unpaired *t*-test, or two-way ANOVA using Graph Pad Prism 7.04 (Graph Pad Software Incorporation, San Diego, CA, USA). Data were expressed as the mean ± SEM for each group. A statistical significance was determined by *p* < 0.05.

## 5. Conclusions

As proposed in Figure 5, challenge exposure to nicotine stimulates α7 nAChRs in the glutamate terminals, increases glutamate release in the CPu, and stimulates mGluR5 located in the GABAergic neurons. Stimulation of mGluR5 linked to Gαq increases the concentration of IP_3_ via elevation of intracellular Ca^2+^ mobilization. This results in the increase of nNOS phosphorylation (activation) followed by NO synthesis. Elevated NO concentration in the synapses of GABAergic neurons activates MMP2 from the inactive form, leading to an increase in BDNF synthesis. Elevation of BDNF then stimulates TrkB in GABAergic neurons, which exerts functions on hyper-glutamatergic neurotransmission caused by challenge exposure to nicotine. In conclusion, elevation of NO synthesis after challenge exposure to nicotine activates MMP2, and upregulates BDNF synthesis through mGluR5-linked IP_3_ pathways in the CPu of rats. This finding may imply the potential role of NO in the upregulation of BDNF under hyper-glutamatergic neurotransmission caused by challenge exposure to nicotine.

## Figures and Tables

**Figure 1 ijms-23-10950-f001:**
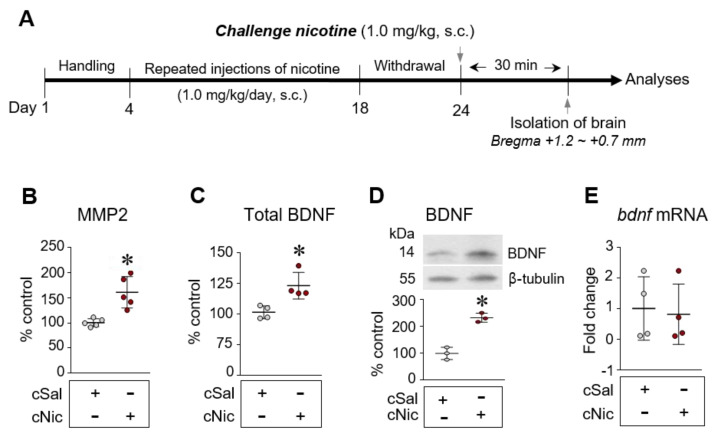
Challenge nicotine increases in MMP 2 activity (**B**) as well as pro- and mature BDNF (total BDNF) (**C**) and BDNF (**D**), but not *bdnf* mRNA (**E**), expression in the CPu of rats. Timeline (**A**). cSal, challenge saline; cNic, challenge nicotine. s.c., subcutaneous injection. *n* = 3–5 per group. * *p* < 0.05 vs. challenge saline group.

**Figure 2 ijms-23-10950-f002:**
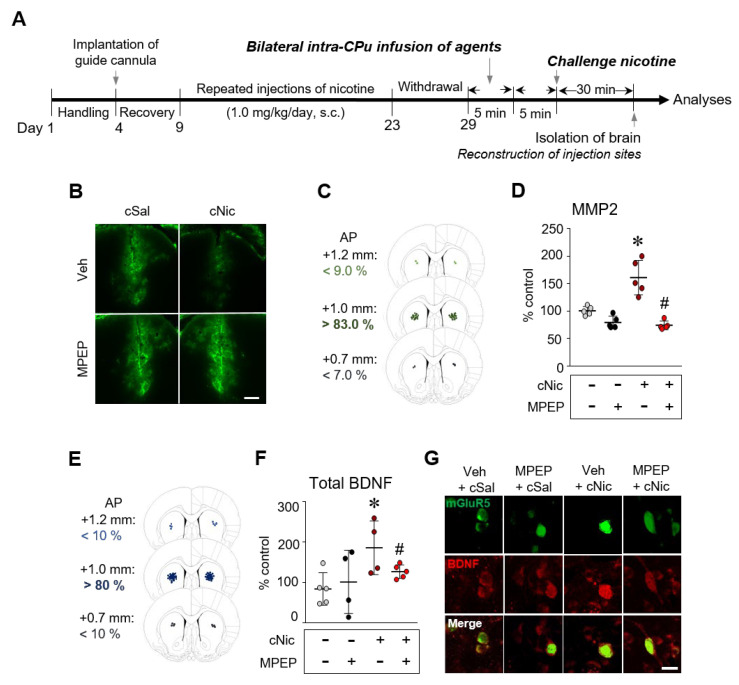
Blockade of mGluR5 decreases the challenge nicotine-induced increases in MMP 2 activity (**D**) and BDNF expression (**F**,**G**) in the CPu. Timeline (**A**). Pretreatment of MPEP does not reduce the challenge nicotine-induced FITC-conjugated fluorescence intensity when compared to the challenge nicotine group (**B**). Reconstruction of the bilateral intra-CPu injection of MPEP prior to challenge nicotine (**C**,**E**). Pretreatment of MPEP decreases the challenge nicotine-induced increase in mGluR5/BDNF-double positive cells in the CPu (*n* = 18–20 slides of 3–4 rats) (**G**). *n* = 4–5 per group. * *p* < 0.05 vs. challenge saline group or challenge saline + vehicle group; # *p* < 0.05 vs. challenge nicotine + vehicle group.

**Figure 3 ijms-23-10950-f003:**
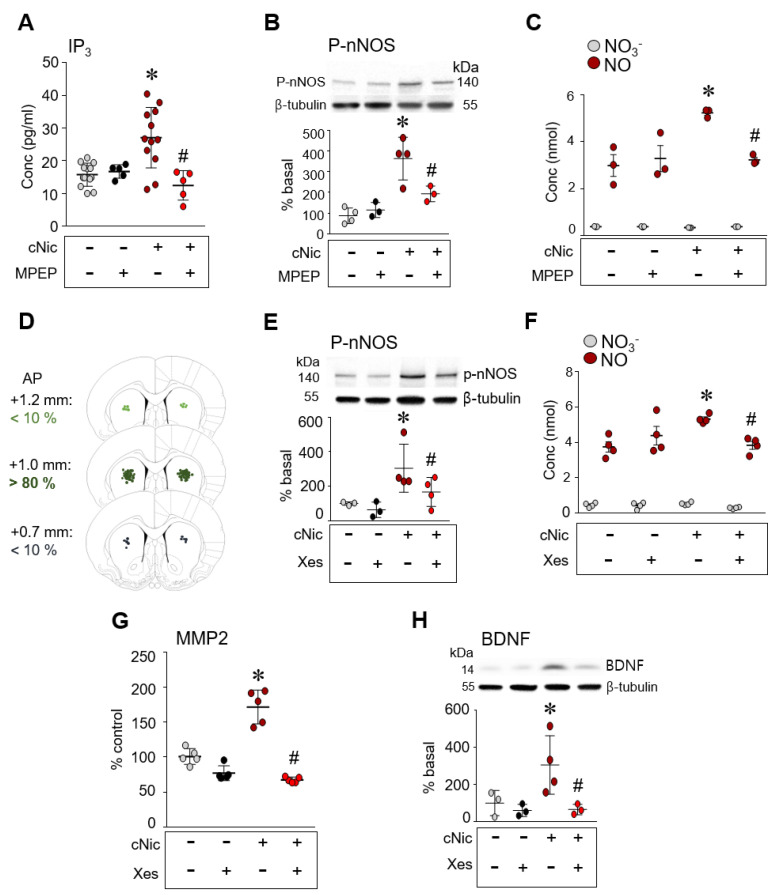
Blockade of mGluR5 and IP_3_ decreases the challenge nicotine-induced increases in IP_3_ (**A**), p-nNOS (**B**,**E**), NO (**C**,**F**), MMP2 activity (**G**), and BDNF (**H**) in the CPu. Reconstruction of the bilateral intra-CPu infusion of xestospongin C (Xes) prior to challenge nicotine (**D**). *n* = 3–12 per group. * *p* < 0.05 vs. challenge saline + vehicle group; # *p* < 0.05 vs. challenge nicotine + vehicle group.

**Figure 4 ijms-23-10950-f004:**
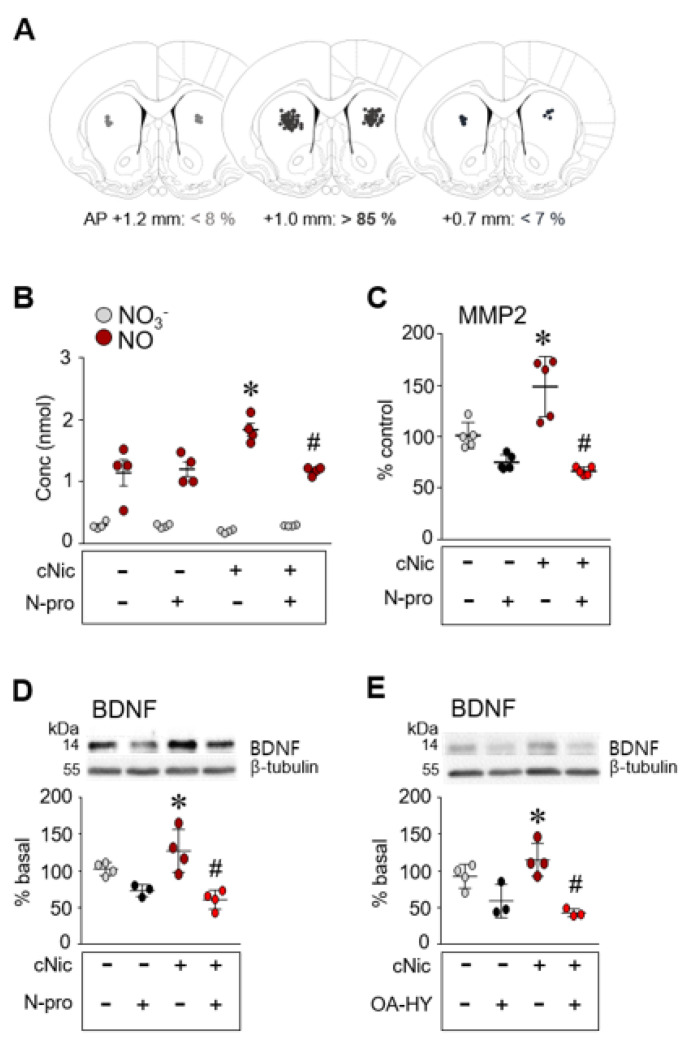
Inhibition of either nNOS or MMP2 decreases the challenge nicotine-induced increases in NO (**B**), MMP2 activity (**C**), and BDNF (**D**,**E**) in the CPu. Reconstruction of the bilateral intra-CPu infusion of Nω-propyl (N-pro) or OA-HY prior to challenge nicotine (**A**). *n* = 3–5 per group. * *p* < 0.05 vs. challenge saline + vehicle group; # *p* < 0.05 vs. challenge nicotine + vehicle group.

**Figure 5 ijms-23-10950-f005:**
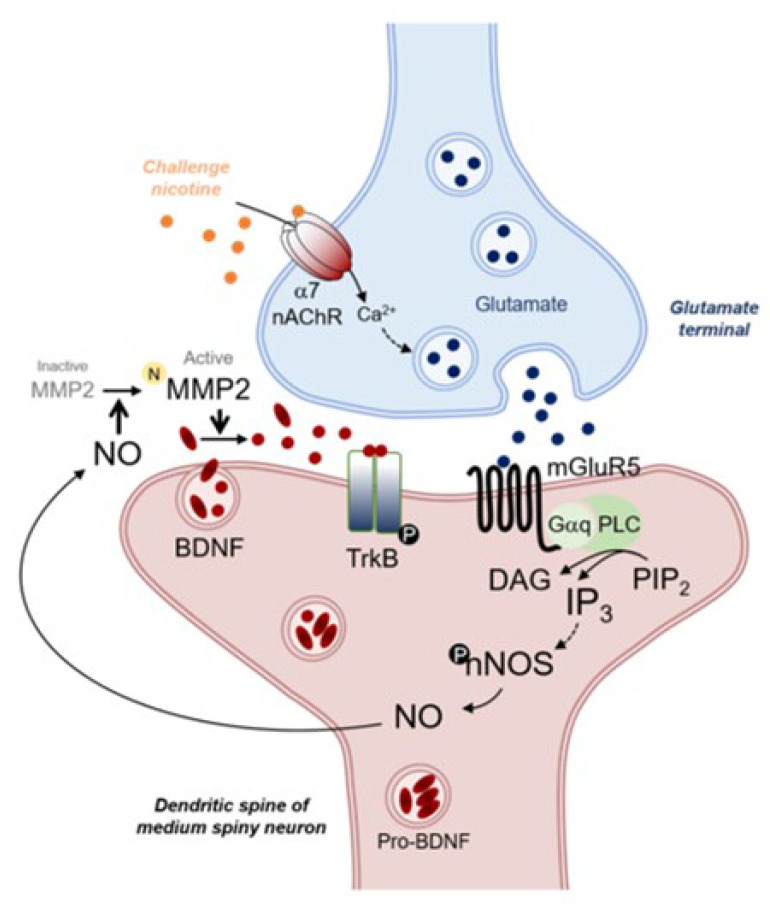
A schematic diagram that proposed the challenge nicotine-induced increase in BDNF synthesis by activating MMP2 via mGluR5-linked NO synthesis in the CPu of rats. Challenge exposure to nicotine increases glutamate release by stimulating α7 nAChR in the glutamate terminal of the CPu, which results in the elevation of nNOS phosphorylation followed by NO synthesis by stimulating mGluR5. This increase in NO concentration by upregulating IP_3_ concentration that is coupled to Gαq further upregulates MMP2 catalytic activity, leading to an increase the synthesis of BDNF in the synapse of a medium spiny neuron. The possible interactions occurred in the GABAergic neuron of the CPu after challenge exposure to nicotine are described in the discussion. Solid and broken arrows represent the direct and indirect stimulation of downstream molecules, respectively. N, nitrosylation; P, phosphorylation; filled red circle, mature BDNF.

## Data Availability

All data generated and/or analyzed during this study are included in this published article.

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
