# Peer review of "Nitric Oxide Linked to mGluR5 Upregulates BDNF Synthesis by Activating MMP2 in the Caudate and Putamen after Challenge Exposure to Nicotine in Rats"

_ijms, 2022, doi:10.3390/ijms231810950_

Round 1

Reviewer 1 Report

The manuscript submitted by Jieun Kim and coauthors is devoted to the effects of nicotine exposure to BDNF synthesis in caudate and putamen. The studying of nicotine effects on brain cells is an actual problem, since millions people around the world actively consume nicotine in everyday life. I have only some comments for the authors.

The authors provide a scheme explaining the observed effects by the activity of neurons. It is known, that glial cells also express mGluRs, nicotine receptors and can release different gliotransmitters and cytokines in Ca2+-dependent manner. According to the literature data, caudate and putamen contain glial cells, however the contribution of these cells in the observed effects is not discussed in the work. Please, discuss it.

Minor points

- "Nitric oxide (NO) coupled to glutamate receptors". This phrase seems to be confusing since "couple" is more appropriate for proteins, such as G-proteins, arrestins etc.

- It is better to change "NO efflux" to "NO synthesis/production".

- "GABA neuron". What do the authors mean? GABAergic neurons?

- Page 3, line 94:"deceased FITC-conjugated fluorescence intensity". Please, explain correctly what was conjugated with FITC.

- Please, explain what "total BDNF" means.

Author Response

Reviewer 1

The manuscript submitted by Jieun Kim and coauthors is devoted to the effects of nicotine exposure to BDNF synthesis in caudate and putamen. The studying of nicotine effects on brain cells is an actual problem, since millions people around the world actively consume nicotine in everyday life. I have only some comments for the authors.

The authors provide a scheme explaining the observed effects by the activity of neurons. It is known, that glial cells also express mGluRs, nicotine receptors and can release different gliotransmitters and cytokines in Ca2+-dependent manner. According to the literature data, caudate and putamen contain glial cells, however the contribution of these cells in the observed effects is not discussed in the work. Please, discuss it.

Response: We modified the second paragraph on page 7 by adding a new sentence concerning the possibility that stimulation of mGluR5 in astrocytes can contribute to the increase in BDNF synthesis by increasing glutamate release in the CPu after challenge nicotine as the reviewer pointed out. 

Minor points

1. "Nitric oxide (NO) coupled to glutamate receptors". This phrase seems to be confusing since "couple" is more appropriate for proteins, such as G-proteins, arrestins etc.

Response: We changed it to ‘linked’ throughout the revised manuscript.

2. It is better to change "NO efflux" to "NO synthesis/production".

Response: We changed it to ‘NO synthesis’ throughout the revised manuscript.

3. "GABA neuron". What do the authors mean? GABAergic neurons?

Response: We changed it to ‘GABAergic neurons’ throughout the revised manuscript to avoid confusion.

4. Page 3, line 94:"deceased FITC-conjugated fluorescence intensity". Please, explain correctly what was conjugated with FITC.

Response: As the reviewer knows, gelatin is conjugated with FITC. We therefore corrected it to ‘gelatin-conjugated fluorescein isothiocyanate (FITC)’ on page 3 of the revised manuscript. 

5. Please, explain what "total BDNF" means.

Response: Total BDNF indicates both pro- and mature-BDNF. Since ELISA applied in this study detects both of them, we phrased it as ‘pro- and mature-BDNF (total BDNF) in figure legend 1 on page 3.  

Reviewer 2 Report

Kim and colleagues determined that the increase in NO concentration after nicotine exposure by IP3 upregulation that is coupled to Gαq, further increases the catalytic activity of MMP2, leading to elevation of BDNF synthesis in the GABA neuron in the caudate and putamen (CPu). The authors have covered their bases very well with these experiments. 

It would be better if the authors could briefly explain in the discussion about the significance of this work and how it is important compared to the other relevant findings.

Author Response

Reviewer 2

Kim and colleagues determined that the increase in NO concentration after nicotine exposure by IP3 upregulation that is coupled to Gαq, further increases the catalytic activity of MMP2, leading to elevation of BDNF synthesis in the GABA neuron in the caudate and putamen (CPu). The authors have covered their bases very well with these experiments. 

It would be better if the authors could briefly explain in the discussion about the significance of this work and how it is important compared to the other relevant findings.

Response: We appreciate the reviewer’s insightful comments. A novel finding obtained from the present study may provide understanding on the potential role of NO in the upregulation of BDNF under hyper-glutamatergic neurotransmission after challenge exposure to nicotine. We therefore described it in the original manuscript at the end of the conclusions on page 10, which is highlighted in blue.

Round 2

Reviewer 1 Report

All my comments have been addressed.